# German validation of three ethics questionnaires: Consequentialist scale, ethical standards of judgment questionnaire, and revised ethics position questionnaire

Birgit Teichmann[1]*, Florian Melchior[1], George Kosteletos[2,3,4]

1 Network Aging Research (NAR), Heidelberg University, Heidelberg, Germany, 2 Applied Philosophy Research Laboratory, National and Kapodistrian University of Athens, Athens, Greece, 3 University Mental Health, Neurosciences and Precision Medicine Research Institute "COSTAS STEFANIS" (UMHRI), Athens, Greece, 4 Open University of Cyprus, Nicosia, Cyprus

* teichmann@nar.uni-heidelberg.de

## Abstract

### Introduction

The Consequentialist Scale (CS) and the Ethical Standards of Judgment Questionnaire (ESJQ) are instruments developed to evaluate the extent of moral reasoning in relation to the two pivotal factors that appear to influence moral decision-making: the degree of harm or benefit produced by the action in question and the consistency of the action with moral norms. In other words, they assess the propensity to utilitarian versus deontological moral reasoning. In contrast, the Ethical Position Questionnaire (EPQ-5) conceptualizes ethical idealism and ethical relativism as meaning-independent, orthogonal dimensions. This study aimed to assess the psychometric properties of German versions of the three mentioned scales in a sample of native German speakers.

### Methods

A convenience sample of 263 participants completed the online survey. Analyses included internal consistency, structural validity, construct validity through the known-groups method, retest-reliability with a subgroup of $n = 102$, and floor and ceiling effects. This study used the STROBE checklist.

### Results

The CS and EPQ-5 showed strong psychometric properties without any noticeable weaknesses. In contrast, the ESJQ displayed significant shortcomings across all analyses, with low internal consistency and poor results in both item analysis and confirmatory factor analysis. The results indicated that deontology, formalism, and idealism were positively correlated with age, while only idealism correlated

**Data availability statement:** The datasets generated during the current study are available in the Open Science Framework (OSF) repository (https://osf.io/bjqfx/).

**Funding:** The author(s) received no specific funding for this work.

**Competing interests:** The authors have declared that no competing interests exist.

significantly with gender, with females scoring higher on the idealism scale. A positive correlation was observed between deontology and formalism with religiosity. With regard to personality, deontology and idealism demonstrated a positive correlation with conscientiousness, whereas utilitarianism exhibited a negative correlation with conscientiousness. A positive correlation between consequentialism and openness was also identified, while a negative correlation between formalism and agreeableness was evident.

## Conclusion

The German versions of the CS and EPQ-5 are reliable and valid instruments for measuring the propensity toward utilitarian and deontological approaches, as well as ethical idealism and relativism. The scales, therefore, serve as invaluable tools for research, training, and professional practice, facilitating comprehension of the aspects of conscious reflection on ethical dilemmas in practice and of responsible action. The ESJQ, however, did not perform well psychometrically in the German translation, as its internal consistency is questionable.

### Introduction

Morality can be defined as a body of moral standards, principles, and values that regulate interpersonal behavior within a society and are accepted as binding by that society [1]. As a construct influenced by cultural and social factors, it is to be expected that there will be variations in the manifestation of morality across diverse geographical regions and at different points in historical time.

In contrast, ethics is the field of philosophical inquiry concerned with the justification of moral values and norms. Various ethical or moral theories can be distinguished from one another on the basis of their distinctive approaches [2]. Some philosophers posit that all ethics can be classified into one of two categories: consequentialism and deontology [3]. The term "consequentialism" was first introduced by Elisabeth Anscombe in 1958 and describes an ethical theory that measures the moral value of an action based on the expected consequences [4]. Consequentialist ethics thus diverge from deontological ethics, which assess the morality of actions based on their intrinsic value rather than their consequences [5]. The definitional details of consequentialism and deontology remain a topic of ongoing debate. The dividing line between the two schools of thought is subject to interpretation and varies according to the perspective. In most of its manifestations, consequentialism, like deontology, draws on fixed principles and intrinsic values. However, it accomplishes so in a manner that is distinct from that of deontology, which has universally valid principles. Hence, consequentialism allows for more flexibility in its practical application, thereby better accounting for the diverse contexts and developments in moral evaluation across cultures [6]. It is worth noting, however, that the traditional and more widely accepted philosophical distinction is that between "deontology" and "utilitarianism", the latter being a form of consequentialist ethical reflection – in the

sense that it focuses on the consequences of an action or rule, but with the characteristic aim of producing the "greatest good for the greatest number" [7]. Furthermore, from an epistemological point of view, it is worth mentioning that virtue ethics represents a third ethical approach alongside utilitarianism and deontology. This approach draws upon the philosophical tradition of Aristotle and emphasizes the concepts of "virtue", "practical wisdom", and the formation of moral character [8].

There has long been an interest in investigating the psychological mechanisms underlying the propensity towards utilitarian approaches to decision-making [9]. In order to achieve this, vignettes containing moral dilemmas have been created; however, these have undergone numerous unexplained alterations over time [10]. Furthermore, the methodology is highly susceptible to criticism, as it employs dilemmas to exemplify a moral principle with the aim of investigating the presence or absence of a given moral principle, which is, in itself, a circular process, and the judgments are solely made in relation to a specific scenario. As a result, it remains unclear what beliefs and values have led to these judgments [11]. Other points of criticism are that the scenarios often depicted present dilemmas that are not representative of real-life experiences, such as the trolley problem [12], or that scenes from very specific areas were used, such as business [13] or sports [14]. Besides, these scenarios result in a number of potential problems for the accurate and reliable measurement. Not only can the context of a scenario influence participants' responses to such scenarios, but the psychometric properties of such scales tend to be less strong than more traditional, item-based assessments [15] and are deemed to lack both ecological validity and generalizability in relation to the examination of moral decision-making [11].

Several scales have been developed with the aim of analyzing moral reasoning in terms of the two key factors that seem to play a role in moral decision-making: the degree of harm or benefit produced by the action in question and the consistency of the action with moral norms, independent of specific behaviors. Two of these are the Consequentialist Scale (CS), developed by Robinson in 2012 [11], and the Ethical Standards of Judgment Questionnaire (ESJQ), developed by Love et al. in 2020 [3]. The CS is a ten-item scale that was created to circumvent the complications and confounding factors inherent to the use of traditional moral dilemmas. The instrument has previously been employed to investigate the influence of ethical attitudes on the propensity to pay for the improvement of farm animals in Germany [16]. However, it has not been validated in German. Furthermore, the scale has already been translated and validated in the Greek language [17].

The ESJQ was developed with the objective of combining the multidimensionality of Brady and Wheeler's (1996) [18] "Measure of Ethical Viewpoints" (MEV) with the conceptual validity of Brady's (1990) [19] "Survey of Ethical Heoetic Aptitudes" (SETA). The ESJQ is a 12-item scale comprising two factors: consequentialism and formalism. Each factor consists of six items, and it is recommended that these be assessed as independent constructs rather than as a single underlying factor. To the best of our knowledge, the ESJQ has not been translated into any other language.

One of the most frequently used instruments is the Ethics Position Questionnaire (EPQ), which postulates that individuals' responses to morally contentious scenarios can be attributed to discrepancies in their intuitive personal moral philosophies [20]. The theory conceptualizes ethical idealism (belief in categorical ideals) and ethical relativism (rejection of universal principles) as meaning-independent, orthogonal dimensions. This allows for the representation of four basic ethical positions: exceptionalism (low idealism and low relativism), subjectivism (low idealism and high relativism), absolutism (high idealism and low relativism), and situationism (high idealism and high relativism) [21,22]. As posited by Forsyth (1980) [22], those who espouse an exceptionist stance adhere to universal moral principles, yet are amenable to the possibility of exceptions contingent on the anticipated consequences. Subjectivists base their decisions on personal feelings and intuition, and thus reject the universality of moral rules. In contrast, absolutists base their actions on inviolable moral norms that apply regardless of possible consequences. Exceptionists, however, reject universal moral rules and evaluate the respective individual situation, but the consequences of their decisions should be justifiable according to moral principles [21]. Thus, according to Forsyth, absolutists present a theoretical predisposition that is analogous to what philosophers would term "deontology", whereas exceptionists present a theoretical predisposition that is analogous

to what philosophers would term "utilitarianism" [22]. Nevertheless, the EPQ is only tangentially and indirectly associated with utilitarianist and deontological frameworks of moral ideology. Although the EPQ has been demonstrated to be a valuable tool for predicting differences in moral judgment across various contexts, some researchers have been unable to validate the two-factor structure, reporting low factor loading or only weak correlations with total scores [23,24]. Furthermore, the correlation between the two scales and associated constructs, such as moral values, has not consistently aligned with the theoretical prediction or joint interaction of the two indices. Consequently, a condensed version, the EPQ-5 scale, was devised that has exhibited convergent and divergent validity and is conceptually consistent with the ethics position theory [24].

As the majority of the questionnaires have been utilized in a multitude of studies within the field of English-language studies, there are currently no validated scales available in German. While the EPQ has already been translated into German, the study by Strack and Gennerich (2007) was conducted with a very small sample size (N = 132). However, questionnaires translated and validated in German are necessary to investigate cultural differences in moral evaluation and to correlate these tendencies with other characteristics, including personality traits, religiosity, and age.

The principal objective of the present study is to translate and validate the CS in German. To ascertain the discriminant validity, the EPQ-5 and the ESQJ were also translated into German, and the structural validity, the construct validity, item analysis, floor and ceiling effects, and retest reliability of all scales were evaluated. As numerous studies have demonstrated that personality traits [25–27], religiosity [28–32], and age [17,33,34] exert an influence on moral ideologies or moral judgments, the following hypotheses were examined:

Older adults tend to exhibit a preference of deontology and idealism, while younger individuals lean more toward utilitarianism and relativism. Those higher in conscientiousness are more sensitive to moral norms than those lower in conscientiousness. Those higher in openness are more sensitive to consequences and moral norms than those lower in openness. Those who express greater religiosity will make less utilitarian judgments.

## Materials and methods

A convenience sample was recruited in Germany between April 10, 2024, and July 26, 2024, through flyers and newsletters, and by forwarding the call to participate in the study via social media such as WhatsApp and Facebook.

The final sample size was N = 263. After four weeks, some of the participants completed the questionnaire a second time. This resulted in a sub-sample of $n$ = 102.

The present study followed the EQUATOR guidelines for reporting research using the "Strengthening the Reporting of Observational Studies in Epidemiology" (STROBE) checklist [35] (S1 Appendix).

### Questionnaire design

Google Forms were used to collect the data, which required participants to answer sequential questions about socio-demographic information, religiosity, previous experience with philosophy, a short version of the Big Five Inventory (BFI-10) [36], the CS [11], the ESJQ [3], and the revised EPQ-5 [22]. At the end of the first questionnaire, respondents were given the option to participate a second time after four weeks. They could create a code to match the data from two surveys and provide their email addresses to receive a reminder to participate the second time. The time needed to complete the questionnaire was estimated to be around 10–20 minutes.

### Big Five Inventory-10 (BFI-10)

The BFI-10 is the shortened German version of the Big Five Inventory by John et al. (1991) [37], which was published and validated by Rammstedt et al. in 2007 [38]. The questionnaire is based on the five-factor model of personality [39] and includes the dimensions of openness to experience, conscientiousness, extraversion, agreeableness, and neuroticism. Each dimension is measured by one item with a positive and one with a negative polarization. Items 1, 3, 4, 5, and 7 are

negatively polarized. A five-point rating scale, ranging from "strongly disagree" (value 1) to "strongly agree" (value 5), is provided for respondents to indicate their level of agreement with the statements presented. The short form, with 10 items, is significantly more economical than other measurement instruments, such as the NEO-FFI with 60 items [40], and has been extensively validated [38,41].

## Consequentialist Scale (CS)

The CS, created by Jeffrey S. Robinson in 2012 [11], measures a person's deontological and consequentialist perspectives and how they influence their moral judgments. The questionnaire comprises ten items, with five items each dedicated to deontology and consequentialism. Participants rate statements such as "Some rules should never be broken" on a 5-point Likert scale from "completely disagree" to "completely agree". The scores for each of the factors reflect the degree to which a participant adheres to a deontological or consequentialist way of thinking. The deontological scale achieved a Cronbach's alpha of 0.69, and the consequentialist beliefs had an alpha of 0.80 in Robinson's original validation (2012) [11].

## Ethical Standards of Judgement Questionnaire (ESJQ)

The ESJQ is another questionnaire designed to measure formalism and consequentialism. Published and validated by Love et al. in 2020 [3], this questionnaire consists of 12 items, six items for each factor, consequentialism and formalism, including statements such as "When people disagree over ethical matters, I strive for workable compromises". Participants indicated their level of agreement with each statement on a 5-point Likert scale ranging from "strongly disagree" to "strongly agree".

## Ethics Position Questionnaire (EPQ-5)

The EPQ-5 is a 10-item questionnaire by O'Boyle and Forsyth (2021) [24], developed to assess idealism and relativism. It is derived from the original EPQ by Forsyth from 1980 [22] and was released in a condensed and enhanced form in 2021. Participants are presented with ten statements, such as "If an action could harm an innocent other, then it should not be done"" which they rate on a 5-point Likert scale from "strongly agree" to "strongly disagree". In order to assess relativism, five items have been devised, in addition to five items for individualism. The questionnaire can be further divided into four sub-scales depending on an individual's scores of relativism and idealism, described in O'Boyle and Forsyth 2021 [24]: exceptionist, absolutist, subjectivist, and situationist.

## Developing the German version of the questionnaires

We utilized the translation back-translation method [42] to translate the English version of the questionnaire into German. Specifically, two native speakers separately translated the original English version into German. Differences in translation were discussed with the research team to ensure cultural adaptation, and a synthesis of the two translations was produced. The German version was back-translated by two people each, who were either native speakers or translators. The original English version and the back-translated versions were compared for consistency, relevance, and meaning of the content. The final version was administered to three researchers with expertise in ethics to ensure that all items were consistent before the questionnaires were finalized. The translated questionnaires are included in the Supportive Information (S2 Appendix).

## Statistical analysis

The data was analyzed using descriptive and inferential statistical methods with IBM SPSS Statistics Version 27 [43]. The psychometric properties of all questionnaires measuring a psychological construct were evaluated, including internal consistency (Cronbach's alpha), structural validity (Principal Component Analysis, PCA), construct validity

(known-groups method), item analysis, floor and ceiling effects, and retest reliability. Additionally, we evaluated correlations between the questionnaire scales and all relevant variables as well as between the EPQ-5 sub-scales and the questionnaire scales.

Following the approach of O'Boyle and Forsyth (2021) [24] and Neupane et al. (2014) [44], participants were categorized into four groups based on their responses to the EPQ-5 using a normative approach, visualized in Table 1. Participants with idealism and relativism scores above the sample median (4.0 and 3.2, respectively) were categorized as high idealists and high relativists, while those with scores below the median were classified as low idealists and relativists.

## Correlation table

To summarize the relationships between variables, we performed Spearman rank correlations [45] for all relevant variables.

## Cronbach's alpha and retest reliability

To ensure the internal consistency of our measurements, we computed Cronbach's alpha, a metric that gauges the degree of shared variance among items. The generally accepted range for Cronbach's alpha, as recommended, falls between 0.70 and 0.90 [46].

To evaluate the test-retest reliability, we compared data from the entire sample (N = 263) with a sub-sample (n = 102) after a four-week interval. We then calculated the interclass correlation coefficient, which quantifies the similarity between the two sets of surveys. To determine retest reliability, we followed the method outlined by Koo and Li [47] in SPSS. This method employs a two-way mixed effects model, considering the mean of k measurements and absolute agreement.

## Construct validity

To evaluate construct validity, we employed the known-groups method, a technique that distinguishes two groups based on anticipated differences in their scale scores. The study used prior experience with philosophy, participant age, and religiosity as grouping variables.

The sample was then divided into a group with "low" religiosity (those scoring 0–6 on a scale from 0 to 13) and a group with "high" religiosity (those scoring 7–13). For the religion variable, participants were asked to rate how religious they consider themselves, how often they attend religious services, and the extent to which their opinions and decisions are influenced by religion. For age, we grouped "low" age (18–30 years) and "high" age (51+ years).

Consequently, we formulated the following hypotheses:

[1] Individuals in the highly religious group would tend to judge less in a utilitarian or consequentialist way but higher on the deontology or formalism scale.

[2] Older adults tend to exhibit a preference of deontology and idealism, while younger individuals lean more toward utilitarianism and relativism. To assess these hypotheses, we employed the Wilcoxon-Mann-Whitney test (WMW) [48] for

**Table 1. EPQ-5 categorization.**

|  |  | Relativism | |
|---|---|---|---|
|  |  | High | Low |
| Idealism | High | Situationist | Absolutist |
|  | Low | Subjectivist | Exceptionist |

all the questionnaires. Furthermore, we investigated whether previous experience with philosophy influences the scale scores in any way, as an exploratory analysis.

## Power analysis

To ensure sufficient statistical power for our analyses, we conducted a power analysis using G*Power 3.1.9.7 software [49] following Kang's guidelines (2021) [50]. For the hypothesis mentioned earlier, we anticipated an effect size of at least d = 0.5, aiming for a desired power of 1 - β = 0.95 and maintaining a significance level of α = 0.05.

Given our prior studies [51,52], which indicated a slightly over-educated sample due to our recruitment strategy, we adjusted the allocation ratio to 3. Based on G*Power's recommendation, we arrived at a total sample size of N = 244 for the Wilcoxon-Mann-Whitney tests.

## Item analysis

We conducted an item analysis for each questionnaire to assess the item-total correlation for all items. This correlation measures how consistently an individual item's score aligns with the overall scale score, providing valuable insights into each item's contribution to the measurement. Additionally, we examined the inter-item correlation to gauge the strength of relationships between different items.

Typically, item-total correlations and mean inter-item correlations ranging from 0.2 to 0.4 are considered indicative of significant informational contributions to the scale. However, higher correlations do not necessarily imply increased reliability. In fact, excessively high correlations may indicate item redundancy, which can artificially inflate the questionnaire's internal consistency [53–55].

## Floor and ceiling effects

Another crucial consideration is the potential occurrence of ceiling or floor effects. Ceiling effects arise when observations cluster at the highest values, such as achieving a perfect score, while floor effects occur when they cluster at the lowest values. Consequently, the accumulation of data at these extreme values creates a ceiling or floor effect, which can distort the data distribution and introduce bias. This bias can lead to misleading results, particularly in analyses assuming a normal distribution [56].

Though specific thresholds for detecting these effects are not universally standardized, we deemed a ceiling or floor effect to be present if more than 10% of all participants scored at the minimum or maximum level on any questionnaire.

Notably, there were no missing data, as Google Forms only accepted completed records.

## Structural validity

We examined if the three questionnaires retained their structural identity originally proposed by their authors by performing a confirmatory factor analysis (CFA) with the maximum likelihood estimation procedure. The results of the CFA were reported according to the recommendations of Jackson et al. (2009) and Schreiber et al. (2006) [57,58]. The thresholds for CFI and TLI are > 0.95, for RMSEA < 0.06, and for SRMR a value of < 0.08 [57].

## Ethical considerations

The study protocol was approved by the Ethics Committee of the Faculty of Behavioral and Cultural Studies, University of Heidelberg, Germany (AZ Teich 2024 1/1). The study was performed according to the ethical standards outlined in the Declaration of Helsinki. Respondents participated voluntarily in the study after being informed about its aim and subsequently provided their written consent for participation. The General Data Protection Regulation (GDPR) in a research context [59] was respected by ensuring the confidentiality and anonymity of the data.

## Results

### Sociodemographic data

All socio-demographic characteristics are presented in Table 2. The full sample consisted of 263 participants with an average age of 46.9 years (SD = 19.4). The gender distribution was 37.3% male, 61.6% female, and 1.1% diverse. With regard to educational attainment, the majority of participants held advanced degrees, with 35.0% having obtained a master's or diploma, 12.9% a PhD, and 11.8% a bachelor's degree. The mean number of years of education was 17.5 (SD = 3.5). The occupations of the participants were diverse, with the largest groups being students (20.2%), retirees (19.8%), academics (23.2%), and others (24.7%). Most individuals who selected "other" for their occupation were employed in administrative roles, technical professions, or as teachers.

The participants' marital status varied, with 39.9% declaring they were married, 25.9% indicating they were single, 25.9% stating they were in a partnership, and 7.6% reporting being divorced. Nearly half of the participants (45.2%) reported that they had children. In terms of religiosity, 55.5% of participants exhibited a score of 2 or less on a scale from 0 to 13. When the religiosity variable is classified, 88.6% of participants are categorized as having "low religiosity" (0–6) and 11.4% as having "high religiosity" (7–13). The majority of participants (76.4%) reported having prior experience with philosophy/ethics, while only 22.1% had formally studied anything related to philosophy.

A comparison of the full sample and the subgroup reveals that both groups are largely similar. However, the subgroup exhibits an average age that is approximately five years higher, with a slightly higher proportion of retirees and a lower proportion of students relative to the full sample.

### Correlations between variables

Table 3 depicts the correlations between all variables. Deontology showed a positive correlation with conscientiousness, age, and religiosity, while utilitarianism correlated negatively with conscientiousness. Consequentialism correlated positively with openness and years of education and negatively with having children. In contrast, formalism was positively correlated with religiosity, age, and having children but negatively with agreeableness and years of education.

Idealism exhibited positive correlations with conscientiousness, age, years of education, and having children. It was also the only variable significantly correlated with gender, with females scoring higher on the idealism scale. Relativism was found to have a negative correlation only with the number of years of education.

Table 4 details the correlation between the CS and ESJQ sub-scales with the categorization that was established in Table 1. Our sample was categorized into 47 exceptionists, 57 absolutists, 46 subjectivists, and 44 situationists. Some participants obtained scores that fell exactly at the median, preventing them from being assigned to any specific subgroup. The analysis of the correlations among the four EPQ-5 ethical positions revealed that the deontology scale of the CS showed correlations with all four EPQ-5 ethical positions. In contrast, the formalism scale of the ESJQ correlated with the subjectivists and situationists.

### Descriptive statistics and internal consistency

Table 5 presents the descriptive characteristics and internal consistencies. The lowest possible score for all scales is 1, and the highest possible score is 5. The CS achieved moderately good values, whereas the ESJQ demonstrated insufficient internal consistencies. The internal consistency of the EPQ-5 scales was sufficient, with 0.87 for idealism and 0.78 for relativism.

### Retest reliability

Table 6 includes the ICC for all sub-scales with their respective 95%-CI. In the retest, all scales achieved good values. However, the consequentialism scale from the EJSQ and the utilitarianism scale from the CS had slightly lower scores, both falling below 0.8, compared to the other scales.

**Table 2. Participant characteristics of the total sample and the subgroup.**

| Characteristics | Full sample (N = 263) n | % | Subgroup (n = 102) n | % |
|---|---|---|---|---|
| **Age** | | | | |
| Mean | 46.9 | | 52.1 | |
| SD | 19.4 | | 19.4 | |
| **Gender** | | | | |
| Male | 98 | 37.3 | 43 | 42.2 |
| Female | 162 | 61.6 | 58 | 59.9 |
| Diverse | 3 | 1.1 | 1 | 1.0 |
| **Education** | | | | |
| 9 years or less | 2 | 0.8 | 1 | 1.0 |
| 10 years | 7 | 2.7 | 3 | 2.9 |
| 12-13 years | 59 | 22.4 | 18 | 17.7 |
| Vocational training | 36 | 13.6 | 9 | 8.8 |
| Bachelor | 31 | 11.8 | 12 | 11.8 |
| Master/Diploma | 92 | 35.0 | 41 | 40.2 |
| PhD | 34 | 12.9 | 15 | 14.7 |
| Others | 2 | 0.8 | 3 | 2.9 |
| **Years of education** | | | | |
| Mean | 17.5 | | 18.2 | |
| SD | 3.5 | | 3.7 | |
| **Occupation** | | | | |
| School student | 3 | 1.1 | 0 | |
| Student | 53 | 20.2 | 15 | 14.7 |
| Unemployed | 4 | 1.5 | 2 | 2.0 |
| Retiree | 52 | 19.8 | 31 | 30.4 |
| Care profession | 6 | 2.3 | 0 | |
| Therapeutical profession | 13 | 4.9 | 5 | 4.9 |
| Physician | 6 | 2.3 | 1 | 1.0 |
| Academic | 61 | 23.2 | 24 | 23.5 |
| Others | 65 | 24.7 | 24 | 23.5 |
| **Marital status** | | | | |
| Divorced | 20 | 7.6 | 8 | 7.8 |
| In partnership | 68 | 25.9 | 22 | 21.6 |
| Single | 68 | 25.9 | 26 | 25.5 |
| Married | 105 | 39.9 | 46 | 45.1 |
| Widowed or deceased partner | 2 | 0.8 | 0 | |
| **Do you have children?** | | | | |
| yes | 119 | 45.2 | 55 | 53.9 |
| no | 144 | 54.8 | 47 | 46.1 |
| **Religiosity** | | | | |
| Low | 233 | 88.6 | 90 | 88.2 |
| High | 30 | 11.4 | 12 | 11.8 |
| **Do you already have experience with the subject of philosophy/ethics?** | | | | |
| yes | 201 | 76.4 | 84 | 82.4 |
| no | 62 | 23.6 | 18 | 17.6 |

*(Continued)*

**Table 2.** (Continued)

| | Full sample (N = 263) | | Subgroup (n = 102) | |
|---|---|---|---|---|
| **Have you studied anything related to philosophy?** | | | | |
| yes | 58 | 22.1 | 29 | 28.4 |
| no | 205 | 77.9 | 73 | 71.6 |

Low religiosity is a score of 0–6 and high religiosity is a score of 7–13.

**Table 3. Heatmap for all variables.**

| | 1 | 2 | 3 | 4 | 5 | 6 | 7 | 8 | 9 | 10 | 11 | 12 | 13 | 14 | 15 | 16 |
|---|---|---|---|---|---|---|---|---|---|---|---|---|---|---|---|---|
| 1. Deontology | 1 | **-.16** | .01 | **.31** | **.48** | -.05 | **.30** | .03 | **.31** | .06 | **.21** | .11 | -.07 | .00 | **.17** | .02 |
| 2. Utilitarism | **-.16** | 1 | -.03 | **.16** | -.01 | .05 | -.06 | .01 | -.05 | -.01 | .06 | .10 | -.01 | -.03 | **-.14** | .03 |
| 3. Consequentialism | .01 | -.03 | 1 | **-.20** | .07 | .05 | -.12 | **.13** | -.05 | .01 | **-.13** | .11 | -.04 | **.16** | -.03 | .02 |
| 4. Formalism | **.31** | **.16** | **-.20** | 1 | **.25** | .03 | **.29** | **-.18** | **.24** | -.09 | **.22** | -.02 | -.07 | -.08 | .12 | **-.12** |
| 5. Idealism | **.48** | -.01 | .07 | **.25** | 1 | -.04 | **.20** | -.05 | **.16** | **.18** | **.17** | .07 | .03 | .02 | **.16** | .12 |
| 6. Relativism | -.05 | .05 | .05 | .03 | -.04 | 1 | -.12 | **-.21** | -.12 | -.06 | .02 | .01 | -.04 | -.06 | -.10 | -.05 |
| 7. Age | **.30** | -.06 | -.12 | **.29** | **.20** | -.12 | 1 | **.13** | **.30** | .06 | **.63** | **.16** | **-.15** | **-.13** | **.29** | .10 |
| 8. YoE | .03 | .01 | **.13** | **-.18** | -.05 | **-.21** | **.13** | 1 | .06 | .06 | .10 | -.01 | -.01 | .06 | .08 | .04 |
| 9. Religiosity | **.31** | -.05 | -.05 | **.24** | **.16** | -.12 | **.30** | .06 | 1 | .02 | **.20** | **.18** | .02 | .01 | **.20** | .10 |
| 10. Gender | .06 | -.01 | .01 | -.09 | **.18** | -.06 | .06 | .06 | .02 | 1 | .01 | .07 | **.25** | **.17** | **.19** | .07 |
| 11. Children | **.21** | .06 | **-.13** | **.22** | **.17** | .02 | **.63** | .10 | **.20** | .01 | 1 | **.16** | **-.20** | -.12 | **.24** | .06 |
| 12. Extraversion | .11 | .10 | .11 | -.02 | .07 | .01 | **.16** | -.01 | **.18** | .07 | **.16** | 1 | **-.19** | .06 | **.21** | **.13** |
| 13. Neuroticism | -.07 | -.01 | -.04 | -.07 | .03 | -.04 | **-.15** | -.01 | .02 | **.25** | **-.20** | **-.19** | 1 | **.14** | **-.16** | -.07 |
| 14. Openness | .00 | -.03 | **.16** | -.08 | .02 | -.06 | **-.13** | .06 | .01 | **.17** | -.12 | .06 | **.14** | 1 | .07 | **.19** |
| 15. Conscientiousness | **.17** | **-.14** | -.03 | .12 | **.16** | -.10 | **.29** | .08 | **.20** | **.19** | **.24** | **.21** | **-.16** | .07 | 1 | .09 |
| 16. Agreeableness | .02 | .03 | .02 | **-.12** | .12 | -.05 | .10 | .04 | .10 | .07 | .06 | **.13** | -.07 | **.19** | .09 | 1 |

QoE, Years of Education. Gender uses 0 for male and 1 for female. Significant correlations are highlighted in bold.

**Table 4. Correlation for the EPQ-5 ethical positions.**

| | 1 | 2 | 3 | 4 | CS Deont. | CS Util. | ESJQ Formalism | ESJQ Conseq. |
|---|---|---|---|---|---|---|---|---|
| 1. Exceptionist | 1 | -0.25** | -0.22** | -0.21* | -0.28** | 0.09 | -0.11 | -0.05 |
| 2. Absolutist | -0.25** | 1 | -0.24** | -0.24** | 0.22** | -0.10 | 0.06 | 0.05 |
| 3. Subjectivist | -0.22** | -0.24** | 1 | -0.21* | -0.21* | -0.06 | -0.13* | 0.03 |
| 4. Situationist | -0.21* | -0.24** | -0.21* | 1 | 0.17* | 0.04 | 0.13* | 0.03 |

## Construct validity

The known-groups analysis, depicted in Table 7, revealed that only the deontology scale of the CS could distinguish between individuals with and without philosophical experience, young and old age groups, and low and high levels of religiosity. All other scales, with the exception of the formalism scale of the ESJQ, exhibited the capacity to differentiate only one of these three groups. The formalism scale showed no difference concerning religiosity.

**Table 5. Descriptive statistics for all questionnaires.**

| | Mean score (SD) | Range (min; max) | Cronbach's Alpha |
|---|---|---|---|
| **CS** | | | |
| Deontology | 3.46 (0.75) | 1; 5 | 0.651 |
| Utilitarism | 1.88 (0.63) | 1; 4.80 | 0.744 |
| **ESJQ** | | | |
| Consequentialism | 4.02 (0.41) | 2.67; 5 | 0.477 |
| Formalism | 2.41 (0.65) | 1; 4.67 | 0.642 |
| **EPQ-5** | | | |
| Idealism | 3.96 (0.78) | 1; 5 | 0.867 |
| Relativism | 3.12 (0.82) | 1; 5 | 0.783 |

**Table 6. Retest for all scales.**

| | Intraclass Correlation Coefficient | 95% – Confidence interval | |
|---|---|---|---|
| | | Lower bound | Upper bound |
| **CS** | | | |
| Deontology | 0.871 | 0.803 | 0.915 |
| Utilitarism | 0.796 | 0.689 | 0.866 |
| **ESJQ** | | | |
| Consequentialism | 0.758 | 0.632 | 0.841 |
| Formalism | 0.861 | 0.787 | 0.909 |
| **EPQ-5** | | | |
| Idealism | 0.856 | 0.780 | 0.905 |
| Relativism | 0.805 | 0.772 | 0.902 |

**Table 7. Construct validity for all scales.**

| Group (n) | Experience with philosophy | | Age | | Religiosity | | Significance[1] |
|---|---|---|---|---|---|---|---|
| | Yes (197) | No (57) | Low (81) | High (125) | Low (233) | High (30) | |
| | Mean rank | | | | | | |
| **CS** | | | | | | | |
| Deontology | 122.4 | 145.0 | 79.7 | 119.0 | 128.0 | 163.2 | Experience* Age** Religiosity* |
| Utilitarianism | 128.3 | 124.9 | 110.8 | 98.8 | 136.5 | 97.1 | Religiosity* |
| **ESJQ** | | | | | | | |
| Consequentialism | 132.6 | 110.0 | 111.6 | 98.2 | 134.4 | 113.2 | Experience* |
| Formalism | 119.2 | 156.1 | 80.8 | 118.2 | 129.9 | 148.0 | Experience** Age** |
| **EPQ-5** | | | | | | | |
| Idealism | 122.8 | 143.8 | 88.0 | 113.5 | 131.8 | 133.2 | Age* |
| Relativism | 128.5 | 124.2 | 109.9 | 99.4 | 135.8 | 102.3 | Religiosity* |

Age was classified into low (18–30 years) and high (51+ years). Low religiosity is a score of 0–6 and high religiosity is a score of 7–13.

*<0.05.

**<0.001.

[1]his column only highlights the category that displayed a significant difference in values.

## Item analysis and floor and ceiling effects

Full details of the item analysis as well as ceiling and floor effects are displayed in Table 8. The inter-item correlation matrices are provided in the Supportive Information (S3 Appendix). The CS achieved satisfactory values for the item analysis and did not exhibit any excessively high or missing inter-item correlations. However, the ESJQ presented some issues: Item 1 ("When people disagree over ethical matters, I strive for workable compromises"), Item 2 ("When thinking of ethical problems, I try to develop practical, workable alternatives"), and Item 7 ("Solutions to ethical problems are usually black and white") showed minimal correlation with the other items on their scale. Additionally, Item 4 ("Solutions to ethical problems usually are seen as some shade of gray") had a low discriminatory power of 0.169, making it barely representative of the scale. The EPQ-5 showed some very high correlations and discriminatory powers, particularly in the idealism scale, where the average item discriminatory power of 0.695 is well above the acceptable range.

Only one instance of a problematic scale limit was observed. The EPQ-5 had a significant ceiling effect for the idealism scale, where 11.4% of participants achieved a maximum score. This distribution is depicted in Fig 1.

**Table 8. Item analysis for all scales.**

|  | Mean item-total correlation | Mean inter-item correlation | Floor effect | Ceiling effect |
|---|---|---|---|---|
| **CS** |  |  |  |  |
| Deontology | 0.408 | 0.276 | No | No |
| Utilitarism | 0.513 | 0.372 | No | No |
| **ESJQ** |  |  |  |  |
| Consequentialism | 0.245 | 0.139 | No | No |
| Formalism | 0.374 | 0.229 | No | No |
| **EPQ-5** |  |  |  |  |
| Idealism | 0.695 | 0.574 | No | Yes (11.4%) |
| Relativism | 0.560 | 0.412 | No | No |

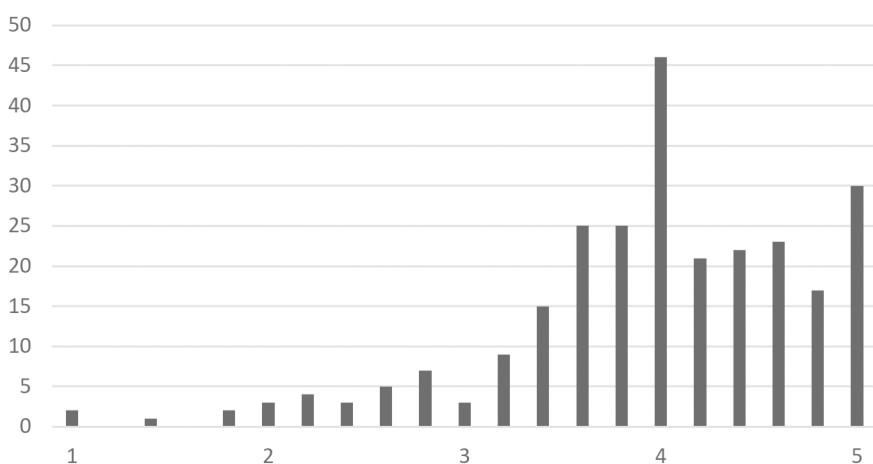

**Fig 1. Distribution of scores for the EPQ-5 idealism subscale.**

**Structural validity**

The confirmatory factor analyses support the previous findings and are displayed in Table 9.

The Chi² tests were significant for all models with p < 0.001, which is expected with a large sample size and should not be considered for model fitting [60,61]. The CS demonstrated acceptable values, though the TLI was slightly low. The ESJQ model could not be validated, as all indices fell well below acceptable levels, with RMSEA > 0.08, CFI and TLI well below 0.95, and SRMR > 0.08 [57]. In contrast, the EPQ-5 achieved excellent values across all fit indices. Standardized estimates for the CS and EPQ-5 are displayed in Figs 2 and 3.

Enhancing the model fit for the CS and the ESJQ would require significant alterations to the structure of the questionnaires.

## Discussion

### Scale properties

The aim of this study was to translate the CS, the ESJQ, and the EPQ-5 and to compare their psychometric properties in a German population. The CS and the EPQ-5 showed strong results across internal consistency, construct validity, retest reliability, and item analysis, with no major weaknesses identified, except for a ceiling effect in the EPQ-5 idealism scale. The EPQ-5 also confirmed its theoretically based structure through confirmatory factor analysis.

In contrast, the ESJQ did not perform as well. The two scales of the ESJQ exhibited poor internal consistency, highlighted by insufficient item correlations, which was evident in the structural equation model.

### Consequentialist scale

To date, the CS has only been translated into Greek by Kosteletos et al. [17] in 2023. Their exploratory factor analysis (EFA) yielded different factors than the original scale. In particular, they identified three factors for their sample, with the consequentialism factor splitting into two factors. In conducting the EFA with two age groups based on a median split, the younger group yielded comparable results to the original study with a two-factor solution, while the older group exhibited almost the same three-factor solution as previously observed. Given that the two-factor solution was based on theoretical principles, we elected to utilize a CFA to substantiate the structural configuration within our sample. Compared with the studies by Robinson 2012 [11], our model demonstrated lower CFI values, but they were still acceptable.

The CS was previously utilized in a study conducted in Germany [16], yet the methodology lacks sufficient detail regarding the translation of the questionnaire into German and the specific psychometric analysis that was conducted. Notwithstanding the assertion that no item exhibited a factor loading below 0.57, the supplementary material discloses that two items have factor loading of 0.222 and 0.272, respectively. Consequently, a comprehensive comparison with our analyses is not feasible. The CS demonstrated superior performance in our sample relative to the Greek sample from Kosteletos et al. (2023) [17]. While the internal consistency was not fully satisfactory, it exhibited higher values than in the

**Table 9. Confirmatory factor analysis for the Consequentialist Scale, the ESJQ and EPQ-5.**

| | Chi | | RMSEA | | | CFI | TLI | SRMR |
|---|---|---|---|---|---|---|---|---|
| | df | p | | Lower bound | Upper bound | | | |
| **CS** | 34 | <0.001 | 0.074 | 0.054 | 0.095 | 0.900 | 0.868 | 0.059 |
| **ESJQ** | 53 | <0.001 | 0.097 | 0.082 | 0.112 | 0.666 | 0.584 | 0.092 |
| **EPQ-5** | 34 | <0.001 | 0.073 | 0.053 | 0.094 | 0.954 | 0.939 | 0.052 |

RMSEA, Root mean square error of approximation with 95% CI; CFI, comparative fit index; TLI, Tucker-Lewis's coefficient; SRMR, standard root mean square residual.

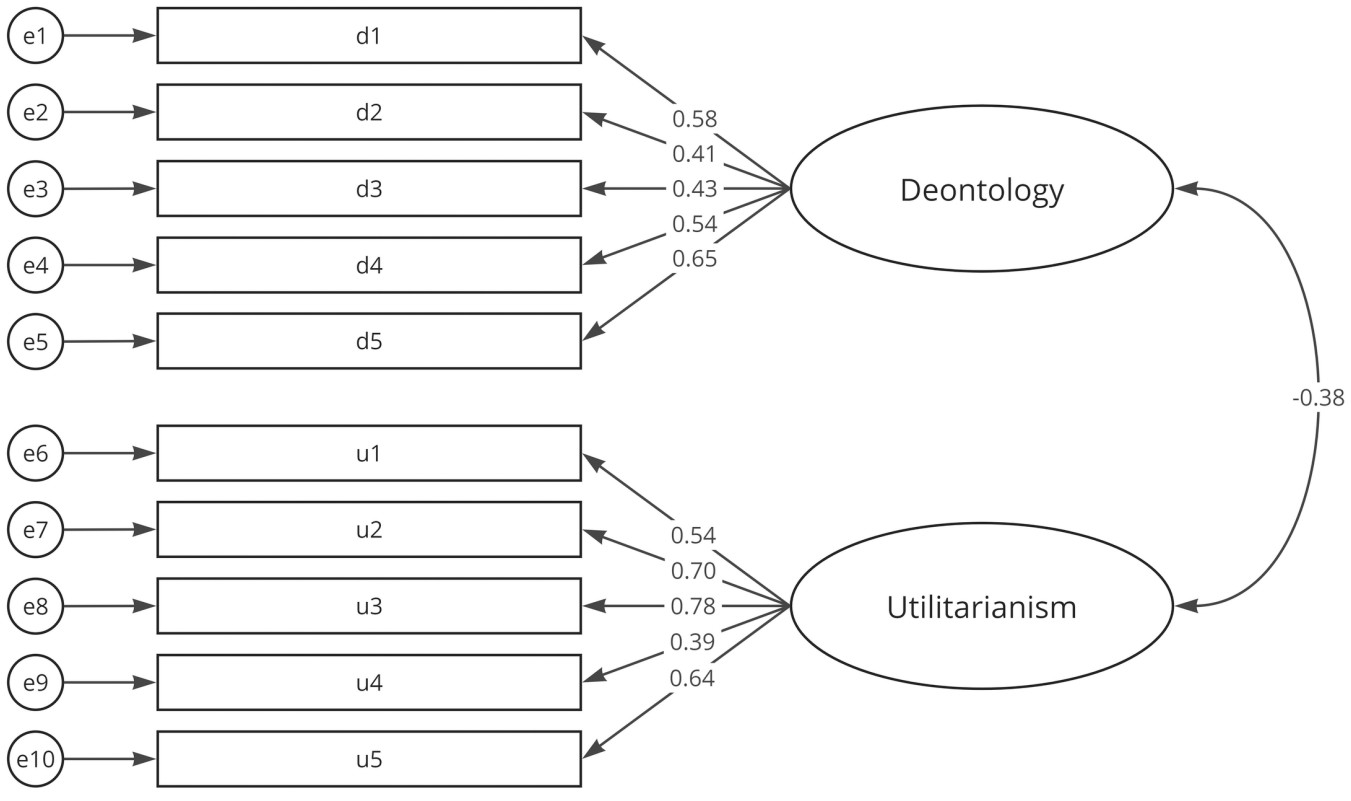

**Fig 2. CFA with standardized estimates for the Consequentialist Scale.**

Greek sample. In both languages, the Cronbach's alpha for consequentialism was higher than for deontology. No other weaknesses in the questionnaire were observed.

**EJSQ**

Notwithstanding the rigorous construction and validation process conducted by Love et al. (2020) [3] for the EJSQ, the resulting German version of the questionnaire yielded unsatisfactory results. In the initial validation, the two sub-scales demonstrated internal consistencies ranging from 0.7 to 0.78, which we hypothesize may contribute to the observed low values. This is due to the fact that the original scale, as proposed by Love et al. (2020) [3], exhibited relatively low internal consistency at its inception. Consequently, it has demonstrated suboptimal performance when applied to disparate samples. All six studies presented in their validation study were carried out with young persons, either students or a sample with a mean range between 32 and 36 years, with only one study where the participants were between 35 and 44 years. As the study population is crucial for the study's internal validity [62,63], it is unsurprising that our results differed. The internal consistency for the consequentialist scale was unacceptable, while for the formalism factor, it was questionable. This low internal consistency explains why we could not find a correlation between the two consequentialism scales of the CS and the ESJQ.

Our findings align with Love et al. (2020) [3] in that the consequentialism sub-scale exhibits markedly higher mean scores in comparison to the formalism sub-scale. However, in contrast to the findings of Love et al. (2020), we identified a significant negative correlation between the two sub-scales. As posited by the authors, formalism and consequentialism are to be regarded as independent constructs, which is why no correlation should be expected [3].

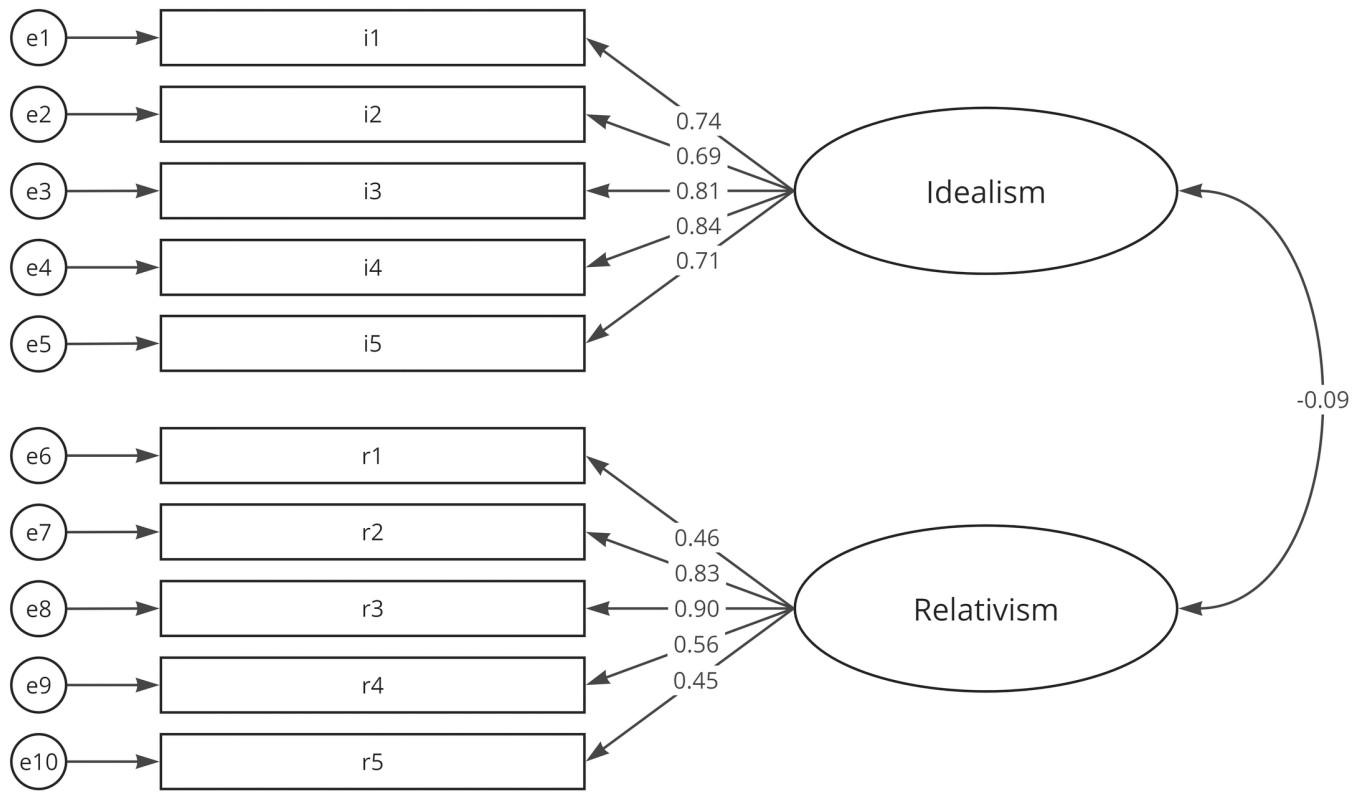

**Fig 3. CFA with standardized estimates for the EPQ-5.**

Since both scales, the CS and the ESJQ, measure the same constructs, correlations between the two subscales, CS-U and ESJQ-C, as well as between the two scales that measure deontology and formalism, CS-D and ESJQ-F, are to be expected. Although a notable positive correlation exists between CS-D and ESJQ-F, no correlation can be identified between CS-U and ESJQ-C.

As the scale has not yet been validated in other samples, comparisons with additional studies are not available at this time. However, we believe that the low correlation between items and the low internal consistency indicates that the items may allow too much room for interpretation or may not correspond well to the constructs they are meant to measure in different cultural contexts.

**EPQ-5**

The internal consistency of both the idealism and relativism scales was found to be comparable to that of the original EPQ-5, which was validated by O'Boyle and Forsyth (2021) [24]. Paschalidou et al. 2024 [64] adapted the 30-item EPQ to align with the specific context of the Greek health and fitness industry, resulting in a shorter form with 12 items. The adapted scale shows variations from the EPQ-5 by having a different item in the idealism scale and adding two items in the relativism scale ("Different types of morality cannot be compared as to rightness" and "Rigidly codifying an ethical position that prevents certain types of action could stand in the way of better human relations and adjustment"). As a result, the relativism scale exhibited higher values.

The CFA confirmed the two-factor solution of the original scale, namely idealism and relativism. The overall model fit assessment indicated satisfactory fit indices that were comparable with the study by O'Boyle and Forsyth (2021) [24] as well as with the study of Paschalidou et al. 2024 [64].

## Impact of age, gender, religiosity, and personality characteristics of moral decision-making

A large number of studies across cultures have shown that age has a strong influence on ethical ideology and ethical judgment [17,34,65,66]. Most studies have found a positive relationship between age and idealism, as older individuals may be more concerned with the welfare of others [66], as well as a positive relationship between age and deontology. Kosteletos et al. (2023) [17] found a significant correlation between age and deontology scores but no correlation between utilitarianism and age. Similarly, we observed a significant correlation between age and deontology and confirmed a significant difference in deontology scores between younger and older participants using the known groups method.

These findings are inconsistent with those presented in a recent article by Lin et al. (2024) [67], who found that older individuals are psychologically inclined to utilitarianism and are less influenced by prevailing norms when making moral decisions. Similar results were reported by Marques and Azevedo-Pereira (2009) [68], who observed that older chartered accountants exhibited a greater tendency toward relativism and proposed that this may be attributed to the notion that ethical standards may decline with the accumulation of experience. Rosen et al. (2016) [69] elucidated the positive effect of age on altruistic moral decisions by emotional empathy in the context of the so-called "positivity effect", which can be observed in older individuals who increasingly attempt to avoid negative affect.

The investigation of gender differences has primarily focused on conventional ethical dilemmas. While Gilligan (1982) [70] posited that men tend to favor a cognitive, abstract, depersonalized approach to moral decision-making, which she termed an "ethic of justice", more recent studies have challenged the notion of significant gender differences with regard to the cognitive component of decision-making. Instead, they suggest that gender differences in maximizing outcomes are relatively small [71]. A meta-analysis conducted by Conway and Gawronski (2013) [72] suggests that gender differences are primarily attributable to an aversion to causing harm. This explanation aligns with our findings, as no correlation was observed between gender and consequentialism/utilitarism, respectively deontology/formalism. Given that our study did not utilize conventional moral dilemmas but instead employed questionnaires that targeted the cognitive rather than the affective aspect, the results are not unexpected. The only statistically significant correlation was between gender and idealism, indicating that women exhibited higher scores on the idealism scale. As idealism entails the rejection of causing harm to others, even when such actions may result in positive outcomes [73], the observed gender difference aligns with the hypothesis that women are less inclined to cause harm.

The relationship between deontology and religious affiliation has been the subject of extensive investigation [32,74–77]. It has been demonstrated that those who make decisions on religious grounds tend to apply rules and principles in their judgments, thus exhibiting deontological rather than consequentialist decision-making [75,76].

This is in line with our results, as our data show a significant correlation between deontology as well as formalism and high religiosity, whereas no correlation between consequentialism and religiosity could be found. Additionally, a notable positive correlation was identified between idealism and religiosity. This is consistent with other studies [31,32]. One limitation of our study is that we did not inquire about the religious denomination of our participants. Even though it can be assumed that most of them may identify as Christians, it is probable that Jews, Muslims, or members of other denominations also took part in the survey. In their study, Love et al. (2020) demonstrated that religious affiliation can exert an influence on judgment. Those who identified as Christian, Muslim, or Hindu were more likely to exhibit a formalistic approach, whereas those who identified as Jewish, atheist, or agnostic, or as belonging to other religions demonstrated a preference for a non-formalistic approach [3]. These findings align with those of Alsaad et al. (2020) [31], who conducted a study with Muslims and identified comparable correlations between idealism and religiosity. They posit that religiosity and idealism are associated with a deeply rooted belief in the well-being of others.

As demonstrated by Kosteletos et al. (2023) [17] in their validation study, pursuing studies in conjunction with philosophy did not exert any discernible impact on moral evaluation. Conversely, our findings revealed a notable divergence in the assessment of the deontology, consequentialism, and formalism scales across individuals with varying degrees of

exposure to philosophical studies. Neither the act of studying nor the engagement with philosophy exerted any discernible influence on the evaluation of the EPQ-5. Although there is not much literature on prior experience in philosophy and moral decision-making, there is some research on the effectiveness of ethical training programs on moral judgment [78–80].

An expanding body of research indicates that discrepancies in moral dilemma judgments may be grounded in fundamental personality characteristics [27]. Our data indicated a positive correlation between deontology and conscientiousness, while utilitarianism exhibited a negative correlation with conscientiousness. As conscientiousness is defined as "the propensity to follow socially prescribed norms for impulse control, to be goal directed, to plan, and to be able to delay gratification" [58], the positive correlation was expected and is in line with the literature [25–27].

The data indicated a positive correlation between consequentialism and openness, which aligns with previous research [25,27] and the theoretical explanation that openness has been linked to complex cognitive functioning [81]. Consequently, moral reflection may facilitate a systematic tendency to evaluate actions in accordance with their consequences.

## Strengths and limitations

Although we made efforts to ensure a diverse sample by including individuals from various social groups and encouraging their social circles to participate in the study, upon comparing our sociodemographic data to the general population in Germany, it appears that our sample is likely to have a higher level of education [82,83].

In the current German population, 33% of individuals aged 25–64 hold a university degree, while in the Eurozone as a whole, it stands at 38.0%, with a steady increase observed since 2014 [84].

Considering that the average age in our sample is 43 and 41 years, respectively, comparing to available statistics for individuals aged 30–34 provides insightful cohort differences and population trends. In the Eurozone, 43.2% of individuals aged 30–34 now hold a university degree, in Germany, it is 37.1% [83]. Notably, over the past decade, the proportion of women with university degrees within a generation in Germany has doubled, and the overall percentage of university graduates continues to rise [85].

However, it is important to note that our sample demonstrates a higher educational attainment compared to the general population, which indicates a potential bias that should be acknowledged.

This bias is a known issue, possibly stemming from the lower participation of less educated individuals in scientific projects. Additionally, the recruitment channels we utilized primarily targeted individuals with a specific interest in research projects. It is also important to mention that those participants who take part again after a four-week interval probably have a higher interest in the topic of ethics, which could lead to a potential bias.

Besides, it is important to consider the limitations associated with the online survey format. Online surveys tend to attract respondents who are technologically proficient or have ample free time [86,87], leading to a potential selection bias and skewed results. Moreover, the absence of personal interaction, as seen in face-to-face interviews, restricts the ability to delve into more detailed or nuanced responses [88]. Lastly, technical difficulties like slow loading times or issues with the survey software can frustrate respondents and potentially impact response rates.

Therefore, we recommend larger sample sizes for future projects and propose an expanded recruitment program. Convenience samples, in general, tend to be statistically biased due to their composition being predominantly WEIRD (Western, Educated, Industrialized, Rich, and Democratic) [89]. Thus, the ability to generalize and make cross-cultural comparisons is limited.

One inherent issue in this study is the potential for respondent fatigue resulting from completing many consecutive scales. The order of these scales was not randomized, since our survey instrument does not support this method. Additionally, the knowledge about genetic technologies was arbitrarily assessed in this study and should instead be measured using a validated questionnaire in future projects.

## Conclusions

The German version of the CS as well as the EPQ-5 are reliable and valid instruments for measuring the propensity to utilize utilitarian and deontological approaches, representing ethical idealism and relativism, respectively. Both demonstrated robust results across internal consistency, construct validity, retest reliability, and item analysis, with no significant shortcomings identified. Furthermore, the theoretically based structures of both instruments were confirmed through CFA. In contrast, the ESJQ has not demonstrated satisfactory psychometric properties in the German translation. Furthermore, the construct validity could not be validated, as all indices fell below the acceptable threshold.

Furthermore, the current study yielded confirmation of the following hypotheses: older adults tend to exhibit a preference for deontology and idealism; those higher in conscientiousness are more sensitive to moral norms than those lower in conscientiousness; and those who express greater religiosity will make less utilitarian judgments.

Ethical decision-making processes are of significant importance in the realms of psychology, sociology, and medicine. However, there is a dearth of instruments that facilitate the investigation of ethical decision-making processes in greater depth. The validated questionnaires could inform health policy by elucidating the foundations of health decisions, whether they are based on moral values and norms or the consequences of an action. Does the act of decision-making itself differ depending on whether the decision is made in a spontaneous manner or after a period of deliberation? Moreover, the questionnaires could be employed to examine the influence of moral values on the decision to be vaccinated or in the utilization of pre-symptomatic diagnostics. This may have implications for psychosocial, sociological and medical practice, as an understanding of ethical decision-making can facilitate the comprehension and interrogation of the ethical attitudes and decision-making processes of professionals. The scales, therefore, serve as invaluable tools for research, training, and professional practice, facilitating comprehension of the aspects of conscious reflection on ethical dilemmas in practice and of responsible action.

## Supporting information

**S1 Appendix.  STROBE Statement.**
(DOCX)

**S2 Appendix.  German translation of the scales.**
(PDF)

**S3 Appendix.  Inter-item correlations.**
(PDF)

## Acknowledgments

This study is independent research. The views expressed in this publication are those of the authors. We would like to thank Ioanna Antigoni Angelidou and Taisiya Baysalova for their contribution to the translation process of the scales as well as the volunteers who participated in the online study. Special thanks to Jason Plaks for permission to translate and validate the Consequentialist Scale into German.

## Author contributions

**Conceptualization:** Birgit Teichmann, George Kosteletos.

**Data curation:** Birgit Teichmann, Florian Melchior.

**Formal analysis:** Birgit Teichmann, Florian Melchior.

**Investigation:** Birgit Teichmann.

**Methodology:** Florian Melchior.

**Project administration:** Birgit Teichmann.

**Resources:** Birgit Teichmann.

**Software:** Birgit Teichmann, Florian Melchior.

**Supervision:** Birgit Teichmann, George Kosteletos.

**Validation:** Birgit Teichmann, Florian Melchior.

**Writing – original draft:** Birgit Teichmann, Florian Melchior.

**Writing – review & editing:** Birgit Teichmann, Florian Melchior, George Kosteletos.

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
