## [Decision Letter · Decision Letter 0]

11 Nov 2024

PONE-D-24-37745The Consequentialist Scale, the Ethical Standards of Judgment Questionnaire, and the Revised Ethics Position Questionnaire: Validation and Comparison of the German VersionsPLOS ONE

Dear Dr. Teichmann,

Thank you for submitting your manuscript to PLOS ONE. After careful consideration, we feel that it has merit but does not fully meet PLOS ONE’s publication criteria as it currently stands. Therefore, we invite you to submit a revised version of the manuscript that addresses the points raised during the review process.

We look forward to receiving your revised manuscript.

Kind regards,

Myriam M. Altamirano-Bustamante

Academic Editor

PLOS ONE

Journal requirements:    When submitting your revision, we need you to address these additional requirements. 1. Please ensure that your manuscript meets PLOS ONE's style requirements, including those for file naming. The PLOS ONE style templates can be found at https://journals.plos.org/plosone/s/file?id=wjVg/PLOSOne_formatting_sample_main_body.pdf and https://journals.plos.org/plosone/s/file?id=ba62/PLOSOne_formatting_sample_title_authors_affiliations.pdf 2. In the online submission form, you indicated that [The data underlying the results presented in the study are available from the authors.]. All PLOS journals now require all data underlying the findings described in their manuscript to be freely available to other researchers, either 1. In a public repository, 2. Within the manuscript itself, or 3. Uploaded as supplementary information.This policy applies to all data except where public deposition would breach compliance with the protocol approved by your research ethics board. If your data cannot be made publicly available for ethical or legal reasons (e.g., public availability would compromise patient privacy), please explain your reasons on resubmission and your exemption request will be escalated for approval.  3. Please include captions for your Supporting Information files at the end of your manuscript, and update any in-text citations to match accordingly. Please see our Supporting Information guidelines for more information: http://journals.plos.org/plosone/s/supporting-information. 

Reviewers' comments:

Reviewer's Responses to Questions

**Comments to the Author**

1. Is the manuscript technically sound, and do the data support the conclusions?

Reviewer #1: Yes

Reviewer #2: Yes

2. Has the statistical analysis been performed appropriately and rigorously? 

Reviewer #1: Yes

Reviewer #2: N/A

3. Have the authors made all data underlying the findings in their manuscript fully available?

Reviewer #1: Yes

Reviewer #2: Yes

4. Is the manuscript presented in an intelligible fashion and written in standard English?

Reviewer #1: Yes

Reviewer #2: Yes

5. Review Comments to the Author

Reviewer #1: The reviewed manuscript is original in nature, I congratulate the authors for the results. All the structures of the article have been preserved. The aim of the work has been clearly defined, the methodology has been properly described, the results are clearly presented. I agree with the authors' suggestion that the study can be conducted among respondents from different socio-economic areas.

Reviewer #2: The present manuscript deals with the validation of the German version of three ethics-inspired scales on the basis of selected psychometric properties. The survey of 263 participants was conducted online using a Google form that only accepted completed forms; 102 participants accepted a second survey four weeks later to assess re-test reliability.

The manuscript has a clear aim, concise methodology and rigorous results. It can therefore be accepted for publication after revision. However, some aspects need to be considered and adjusted. As a reviewer, I try to be systematic in my advice.

Some written mistakes

Line 85: You write “low subjectivism and low relativism”. Do you mean “low idealism and high relativism”?

Line 156: You write “idealism and realism”. Do you mean “idealism and relativism”?

Line 468: You write “religiois”. Do you mean “religious”?

Title

The title is rather long and less synoptic. I suggest an alternative: "German validation of three ethical questionnaires". Or more explicitly: "German validation of three ethical questionnaires: Consequentialist Scale, Ethical Standards of Judgement Questionnaire, and Revised Ethical Position Questionnaire". If accepted by the Editors, abbreviated as CS, EPQ-5 and ESJQ.

Abstract

Objective (perhaps more accurate): The aim is to assess the psychometric properties of the German version of the proposed scales.

In the "Methods" section, I miss the mention of statistical methods.

The "Results" section does not include the associations with age, gender, religiosity and personal characteristics.

In the "Conclusion" section, the possibilities of (clinical) implementation of the analysed questionnaires are not discussed.

Introduction

The authors state in the introduction that they are dealing with a philosophical issue, namely the distinction between a “consequentialist” and a “deontological” approach, and that behavioural styles can be identified in people that can be assigned to one or the other approach. They even suggest that these styles are orthogonal to each other at the level of behaviour. This is a very daring approach, which the EPQ-5 develops theoretically by formulating two dimensions (relativism vs. idealism), each of which is dichotomised into four typologies within a matrix (situationist, absolutist, subjectivist, exceptionist). These typologies are not intuitive and a pure definition according to the matrix (e.g. "absolutist" = low relativism + high idealism) is much too narrow for me. The authors should explain these typologies a little. The authors also distinguish between "morality" and "ethics", which is philosophically legitimate (but without semantic consensus), forgetting that there is a third ethical approach, namely “virtue ethics” (in the tradition of Aristotle and particularly important for medicine, see for example Edmund Pellegrino). I have a great deal of respect for Elisabeth Anscombe, but the traditional philosophical distinction is between "utilitarianism" (by John Stuart Mill and Jeremy Bentham) - not "consequentialism", which means the same thing but is a little more humanistic than “utilitarianism”- and deontological ethics (by Immanuel Kant, with an interesting addition by Georg Simmel - self-legislation, “Selbstgesetzgebung”). I do not know whether the authors had an application to medicine and psychology in mind, as these meta-ethical approaches have not proved successful there, but rather judgement according to ethical principles (e.g. Beauchamp and Childress). A statement on this matter would be helpful.

Method

The recruitment of participants, the description of the scales used and the translation method of the questionnaires are accurate. The subsection 'statistical analysis' is not clearly structured for me: I miss a description of the statistical methods used for each step and the associated statistical parameters. I appreciate the power analysis and the consideration of floor and ceiling effects, but the statistical methods (e.g. construct validity, item analysis, correlation matrix and structural equation modelling) are not systematically explained step by step.

Please: describe each step and the corresponding statistical method, justify the chosen method and describe the significance of the statistical parameters (critical values) for the interpretation (e.g. correlation index, alpha, comparative fit index, etc.).

In psychometric analyses, I consider it important to first justify whether the questionnaire is a formative or a reflective measurement model, because "the use of an incorrect measurement model undermines the content validity of the constructs, misrepresents the structural relationships between them, and ultimately reduces the usefulness of [the underlying theory]" (T. Coltman et al. 2008; but especially Edwards and Bagozzi, 2000). I believe that the implemented scales are reflective (e.g. the indicators of the construct EPQ-5 are considered to be caused by assumed dimensions of the construct); this is because the authors use confirmatory factor analysis and not criterion validity.

The construct validity chosen in this research is not justified. I do not find in the manuscript the statistical method implemented. The dichotomous variables are critical because of the lack of a continuous approach (by accepting loss of information). Why did the authors not use logistic or linear multivariate models? (e.g. Logistic: dichotomous experience of philosophy - a bit strange variable -, dichotomous age, and dichotomous religiosity as dependent variables in each model and other variables as regressors/associates; (e.g. logistic: dichotomous experience of philosophy - a bit strange variable -, dichotomous age and dichotomous religiosity as dependent variables in each model and other variables as regressors/associates; or linear multivariate models: scores of each questionnaire or questionnaire dimensions as dependent continuous variables, three dichotomous variables as continuous variables and other variables as personality dimensions or socio-demographic variables as regressors/associates?

What can the authors say about the convergent or divergent validity of the questionnaires?

In my opinion, the orthogonality of the construct questionnaires has not been assessed (or displayed in the manuscript).

What can the authors say about the representativeness of the subsample (n=102) four weeks later? Usually researchers compare the same subsample or the whole sample and a subsample if the subsample can be considered representative.

Tables

As a general rule, I recommend more detailed captions in the tables to make them self-explanatory.

What do the different colours in Table 2 mean? The table is too large; I suggest reducing on the dimensions of the constructs. Some correlations have not been discussed, e.g. neuroticism and gender, religiosity, and having children.

Table 4: The internal consistency is, in my opinion, too concise. For example, the “average interitem covariance” is statistically relevant. I don't see whether some items can be removed from the model in the German version because the alpha of the item is > alpha of the whole scale. These tables can be included as supplementary material.

Table 6 is not clear. See the comments in the section above.

Table 8 needs to describe critical values of parameters.

Results and Discussion

This research aims to validate the German version of three ethics-based questionnaires. In this respect, the 'Results' section should systematically present the psychometric findings in accordance with the methodological comments further up in the text. The relationships between the assessed scales and factors/dimensions of the constructs with the other measured variables must be explained, focusing on the relevant findings.

In the "Discussion" section, I miss a reflection on the implications of these questionnaires for psychosocial, sociological or medical practice: What is the purpose of these questionnaires? What questions could these questionnaires be used for?

The section on limitations is well argued. Considerations of convergent/divergent validity could be added.

The literature is current, appropriate and detailed.

6. PLOS authors have the option to publish the peer review history of their article (what does this mean? ). If published, this will include your full peer review and any attached files.

**Do you want your identity to be public for this peer review?** For information about this choice, including consent withdrawal, please see our Privacy Policy .

Reviewer #1: **Yes: ** Jolanta Lewko

Reviewer #2: No

---

## [Author Response · Author response to Decision Letter 1]

18 Dec 2024

Ref.: PONE-D-24-37745

Response to Reviewers

Dear Dr. Altamirano-Bustamante,

Thank you for the opportunity to submit a revised version of our manuscript entitled "The Consequentialist Scale, the Ethical Standards of Judgment Questionnaire, and the Revised Ethics Position Questionnaire: Validation and Comparison of the German Versions" for publication in PlosOne. We are grateful for the feedback from you and the reviewers and have made changes to our paper based on your suggestions. These changes are highlighted in the manuscript. Below, in blue, we have provided a detailed response to each of the reviewers' comments and concerns. Thank you for your time and effort in helping us to improve our work.

Reviewer #1: The reviewed manuscript is original in nature, I congratulate the authors for the results. All the structures of the article have been preserved. The aim of the work has been clearly defined, the methodology has been properly described, the results are clearly presented. I agree with the authors' suggestion that the study can be conducted among respondents from different socio-economic areas.

Thank you for your kind words and recognition of our efforts.

Reviewer #2: The present manuscript deals with the validation of the German version of three ethics-inspired scales on the basis of selected psychometric properties. The survey of 263 participants was conducted online using a Google form that only accepted completed forms; 102 participants accepted a second survey four weeks later to assess re-test reliability.

The manuscript has a clear aim, concise methodology and rigorous results. It can therefore be accepted for publication after revision. However, some aspects need to be considered and adjusted. As a reviewer, I try to be systematic in my advice.

Thank you for your kind words and we appreciate your valuable advice!

Some written mistakes

Line 85: You write “low subjectivism and low relativism”. Do you mean “low idealism and high relativism”?

Thank you for this comment!! You are completely right and we changed “low subjectivism and low relativism” to “low idealism and high relativism”.

Line 156: You write “idealism and realism”. Do you mean “idealism and relativism”?

Thank you for bringing this to our attention. We have changed “realism” to “relativism”.

Line 468: You write “religiois”. Do you mean “religious”?

Thank you for bringing this to our attention. We have changed “religiois”. Do you mean “religious”.

Title

The title is rather long and less synoptic. I suggest an alternative: "German validation of three ethical questionnaires". Or more explicitly: "German validation of three ethical questionnaires: Consequentialist Scale, Ethical Standards of Judgement Questionnaire, and Revised Ethical Position Questionnaire". If accepted by the Editors, abbreviated as CS, EPQ-5 and ESJQ.

Thank you for this comment. We agree with changes the title to “German validation of three ethics questionnaires: Consequentialist Scale, Ethical Standards of Judgment Questionnaire, and Revised Ethics Position Questionnaire” and we changed this in the manuscript (line 1-2)

Abstract

Objective (perhaps more accurate): The aim is to assess the psychometric properties of the German version of the proposed scales.

Thank you for your comment, we changed “The objective of this study was to empirically investigate the German translation of the three scales in a sample of native German speakers.” to “This study aimed to assess the psychometric properties of German versions of the three mentioned scales in a sample of native German speakers.”

In the "Methods" section, I miss the mention of statistical methods.

Thank you for your comment. As we explain in detail in the Methods part, analyses included internal consistency, structural validity, construct validity through the known-groups method, retest-reliability with a subgroup of n = 102, and floor and ceiling effects. Everything is mentioned, maybe you want us to mention the statistical tests used, like T-test, factor analysis. This is normally not mentioned in a validation study, as it is more important to explain in detail which validation analyses are done.

The "Results" section does not include the associations with age, gender, religiosity and personal characteristics.

As we wanted to mention only the main results of the study, we mentioned only the results of the psychometric analysis. But as the abstract can be 350 words, we added – as suggested – the associations with age, gender, religiosity and personality: “The results indicated that deontology, formalism and idealism were positively correlated with age, while only idealism correlated significantly with gender, with females scoring higher on the idealism scale. A positive correlation was observed between deontology and formalism with religiosity. With regard to personality, deontology and idealism demonstrated a positive correlation with conscientiousness, whereas utilitarianism exhibited a negative correlation with conscientiousness. A positive correlation between consequentialism and openness was also identified, while a negative correlation between formalism and agreeableness was evident.”

In the "Conclusion" section, the possibilities of (clinical) implementation of the analysed questionnaires are not discussed.

Thank you to mention this. We added the sentence: “The scales, therefore, serve as invaluable tools for research, training, and professional practice, facilitating comprehension of the aspects of conscious reflection on ethical dilemmas in practice and of responsible action.“

Introduction

The authors state in the introduction that they are dealing with a philosophical issue, namely the distinction between a “consequentialist” and a “deontological” approach, and that behavioural styles can be identified in people that can be assigned to one or the other approach. They even suggest that these styles are orthogonal to each other at the level of behaviour. This is a very daring approach, which the EPQ-5 develops theoretically by formulating two dimensions (relativism vs. idealism), each of which is dichotomised into four typologies within a matrix (situationist, absolutist, subjectivist, exceptionist). These typologies are not intuitive and a pure definition according to the matrix (e.g. "absolutist" = low relativism + high idealism) is much too narrow for me. The authors should explain these typologies a little.

We are grateful for your comment. We did not intend to provide an exhaustive account; however, we recognise the value of offering a concise overview to facilitate comprehension. Consequently, we have incorporated a brief description in line 104: “As posited by Forsyth (1980) [1], those who espouse an exceptionist stance adhere to universal moral principles, yet are amenable to the possibility of exceptions contingent on the anticipated consequences. Subjectivists base their decisions on personal feelings and intuition, and thus reject the universality of moral rules. In contrast, absolutists base their actions on inviolable moral norms that apply regardless of possible consequences. Exceptionists, however, reject universal moral rules and evaluate the respective individual situation. Nevertheless, the consequences of their decisions should be justifiable according to moral principles [21]. Thus, according to Forsyth, absolutists present a theoretical predisposition that is analogous to what philosophers would term "deontology", whereas exceptionists present a theoretical predisposition that is analogous to what philosophers would term "utilitarianism" [22].”

The authors also distinguish between "morality" and "ethics", which is philosophically legitimate (but without semantic consensus), forgetting that there is a third ethical approach, namely “virtue ethics” (in the tradition of Aristotle and particularly important for medicine, see for example Edmund Pellegrino). I have a great deal of respect for Elisabeth Anscombe, but the traditional philosophical distinction is between "utilitarianism" (by John Stuart Mill and Jeremy Bentham) - not "consequentialism", which means the same thing but is a little more humanistic than “utilitarianism”- and deontological ethics (by Immanuel Kant, with an interesting addition by Georg Simmel - self-legislation, “Selbstgesetzgebung”). I do not know whether the authors had an application to medicine and psychology in mind, as these meta-ethical approaches have not proved successful there, but rather judgement according to ethical principles (e.g. Beauchamp and Childress). A statement on this matter would be helpful.

Thank you for your comment! It appears that two distinct levels of analysis are being conflated in this discussion. Virtue ethics is not a third option in the distinction between ethics and morality; however, it is a third option in the distinction between deontology and consequentialism.

The terms 'ethics' and 'morality', which are often used synonymously in colloquial language, are clearly distinguished in philosophical discourse, at least in English and German. Ethics is defined as the science of morality. Morality is conceived of as a system of standards that strives to delineate the parameters of acceptable conduct, and thus, it is presumed to be universally applicable. This can be defined in clear terms by reference to the principles or values concerned. However, given that multiple systems of norms can be defined, it follows that a plurality of morals must also exist (religious, political, just).

Virtue ethics, in contrast, may be regarded as a third normative ethical system, alongside consequentialism and deontology. As the three questionnaires under consideration here target only the two categories typically employed by philosophers – consequentialism and deontology – virtue ethics has not been mentioned in the introduction. However, we are happy to do so. We also added a sentence why we used the term “consequentialism” and not utilitarism, as they are not exactly the same:

line 61: “It is worth noting, however, that the traditional and more widely accepted philosophical distinction is that between "deontology" and "utilitarianism", the latter being a form of consequentialist ethical reflection - in the sense that it focuses on the consequences of an action or rule, but with the characteristic aim of producing the "greatest good for the greatest number" [7]. Furthermore, from an epistemological point of view, it is worth mentioning that virtue ethics represents a third ethical approach alongside utilitarianism and deontology. This approach draws upon the philosophical tradition of Aristotle and emphasises the concepts of "virtue", "practical wisdom" and the formation of moral character [8].

Method

The recruitment of participants, the description of the scales used and the translation method of the questionnaires are accurate.

Thank you!

The subsection 'statistical analysis' is not clearly structured for me: I miss a description of the statistical methods used for each step and the associated statistical parameters. I appreciate the power analysis and the consideration of floor and ceiling effects, but the statistical methods (e.g. construct validity, item analysis, correlation matrix and structural equation modelling) are not systematically explained step by step. Please: describe each step and the corresponding statistical method, justify the chosen method and describe the significance of the statistical parameters (critical values) for the interpretation (e.g. correlation index, alpha, comparative fit index, etc.).

Thank you for your comment, we added further clarification for analyses we used in the methods section about the correlation matrix and the CFA with the thresholds we used.

The rest of the statistical methods are described in detail in the sections “Statistical analysis” in the following subsections:

Correlation heatmap; Cronbach’s alpha and retest reliability; Construct validity; Power analysis; Item analysis; Floor and ceiling effects; Structural validity;

In psychometric analyses, I consider it important to first justify whether the questionnaire is a formative or a reflective measurement model, because "the use of an incorrect measurement model undermines the content validity of the constructs, misrepresents the structural relationships between them, and ultimately reduces the usefulness of [the underlying theory]" (T. Coltman et al. 2008; but especially Edwards and Bagozzi, 2000). I believe that the implemented scales are reflective (e.g. the indicators of the construct EPQ-5 are considered to be caused by assumed dimensions of the construct); this is because the authors use confirmatory factor analysis and not criterion validity.

We thank you for the interesting source and agree that it is essential to define the direction of relationships between constructs in a theoretical model. However, in this case, we have not established an underlying theory but instead estimate latent variables by summing the associated items. As you suggested, a valuable step for future research could involve developing a comprehensive theory and exploring the underlying relationships and their directions in greater depth.

The construct validity chosen in this research is not justified. I do not find in the manuscript the statistical method implemented. The dichotomous variables are critical because of the lack of a continuous approach (by accepting loss of information). Why did the authors not use logistic or linear multivariate models? (e.g. Logistic: dichotomous experience of philosophy - a bit strange variable -, dichotomous age, and dichotomous religiosity as dependent variables in each model and other variables as regressors/associates; (e.g. logistic: dichotomous experience of philosophy - a bit strange variable -, dichotomous age and dichotomous religiosity as dependent variables in each model and other variables as regressors/associates; or linear multivariate models: scores of each questionnaire or questionnaire dimensions as dependent continuous variables, three dichotomous variables as continuous variables and other variables as personality dimensions or socio-demographic variables as regressors/associates?

You are absolutely correct that we use various scales to analyze aspects of the questionnaires, and no uniform metric scales are applied. However, there are several reasons for this approach:

We believe that using dichotomous variables does not necessarily mean losing information. For instance, categorizing participants into "low religiosity" and "high religiosity" provides a more robust basis for further analysis compared to using numerical values from 0 to 13. This is because differences, such as between a score of 7 and 8, are difficult to interpret meaningfully, whereas distinguishing between high and low religiosity is more straightforward.

Furthermore, it is important to note that any sociodemographic variable can theoretically be measured in infinite ways and at various levels of depth. However, since we aim to design a concise questionnaire, a few short questions represent the most economical way for us to obtain focused information.

This ties into a crucial final point: ensuring comparability between studies. We achieve this by using the same or similar metrics, enabling direct comparisons across studies. For example, the question regarding previous experience with philosophy is identical to that in the study by Kosteletos et al. (2023), making direct comparisons straightforward and reliable.

What can the authors say about the convergent or divergent validity of the questionnaires?

In my opinion, the orthogonality of the construct questionnaires has not been assessed (or displayed in the manuscript).

Thank you for your comment. As the manuscript is already long, we tried to report only the main results. But you can find at different parts of the discussion some paragraphs regarding convergent validity, as e.g. in line 451: “Since both scales, the CS and the ESJQ, measure the same constructs, correlations between the two subscales, CS-U and ESJQ-C, as well as between the two scales that measure deontology and formalism, CS-D and ESJQ-F, are to be expected. Although a notable positive correlation exists between CS-D and ESJQ-F, n

---

## [Decision Letter · Decision Letter 1]

11 Feb 2025

German validation of three ethics questionnaires: Consequentialist Scale, Ethical Standards of Judgment Questionnaire, and Revised Ethics Position Questionnaire

PONE-D-24-37745R1

Dear Dr. Birgit Teichmann

We’re pleased to inform you that your manuscript has been judged scientifically suitable for publication and will be formally accepted for publication once it meets all outstanding technical requirements.

Kind regards,

Myriam M. Altamirano-Bustamante

Academic Editor

PLOS ONE

Additional Editor Comments (optional):

Reviewers' comments:

Reviewer's Responses to Questions

**Comments to the Author**

1. If the authors have adequately addressed your comments raised in a previous round of review and you feel that this manuscript is now acceptable for publication, you may indicate that here to bypass the “Comments to the Author” section, enter your conflict of interest statement in the “Confidential to Editor” section, and submit your "Accept" recommendation.

Reviewer #1: All comments have been addressed

Reviewer #2: All comments have been addressed

2. Is the manuscript technically sound, and do the data support the conclusions?

Reviewer #1: No

Reviewer #2: Yes

3. Has the statistical analysis been performed appropriately and rigorously? 

Reviewer #1: Yes

Reviewer #2: Yes

4. Have the authors made all data underlying the findings in their manuscript fully available?

Reviewer #1: Yes

Reviewer #2: Yes

5. Is the manuscript presented in an intelligible fashion and written in standard English?

Reviewer #1: Yes

Reviewer #2: Yes

6. Review Comments to the Author

Reviewer #1: Dear authors, the manuscript raises a very interesting topic, congratulations on undertaking research in this area. I believe that the research was conducted reliably and the research results were developed in accordance with the principles of writing original works. The aim of the work was clearly defined, the research tools used were properly characterized. The statistical analysis is correct and the conclusions result from the aim of the research. All comments from the reviewers have been corrected. I recommend the work for publication.

Reviewer #2: Dear authors,

Thank you for considering my comments, which you have implemented well and concisely in the manuscript. The focus of my comments and also your corrections and additions were the meaning of the construct and the refinement of the ethical terminology as well as the methodological part. I understood that you wanted to concentrate on the main results and therefore reduced the breadth of the results. That's fine. You ask me to specify which part of the results are missing. I meant a more systematic linking of the research question, the methods used to answer the questions and the single results. Perhaps I was too meticulous in my desire to systematise the manuscript more strongly throughout; the information has all been presented, important explanations have been included, so I will abstain from further systematisation of details.

From my point of view, the manuscript can be accepted for publication in its current form.

7. PLOS authors have the option to publish the peer review history of their article (what does this mean? ). If published, this will include your full peer review and any attached files.

**Do you want your identity to be public for this peer review?** For information about this choice, including consent withdrawal, please see our Privacy Policy .

Reviewer #1: **Yes: ** Jolanta Lewko

Reviewer #2: No

---

## [Editor Report · Acceptance letter]

PONE-D-24-37745R1

PLOS ONE

Dear Dr. Teichmann,

I'm pleased to inform you that your manuscript has been deemed suitable for publication in PLOS ONE. Congratulations! Your manuscript is now being handed over to our production team.

Kind regards,

on behalf of

Dr. Myriam M. Altamirano-Bustamante

Academic Editor

PLOS ONE